# Deep Learning Improves Prediction of Cardiovascular Disease-Related Mortality and Admission in Patients with Hypertension: Analysis of the Korean National Health Information Database

**DOI:** 10.3390/jcm11226677

**Published:** 2022-11-10

**Authors:** Seung-Jae Lee, Sung-Ho Lee, Hyo-In Choi, Jong-Young Lee, Yong-Whi Jeong, Dae-Ryong Kang, Ki-Chul Sung

**Affiliations:** 1Division of Cardiology, Department of Internal Medicine, Kangbuk Samsung Hospital, Sungkyunkwan University School of Medicine, 29 Saemunan-ro, Jongno-gu, Seoul 03181, Korea; 2Department of Internal Medicine, Yonsei University College of Medicine, 50-1 Yonsei-ro, Seodaemun-gu, Seoul 03722, Korea; 3Department of Information and Statistics, Yonsei University, 20 Ilsan-ro, Wonju 26426, Korea; 4Artificial Intelligence Big Data Medical Center, Yonsei University Wonju College of Medicine, 20 Ilsan-ro, Wonju 26426, Korea

**Keywords:** cardiovascular disease, mortality, hospitalization, hypertension, artificial intelligence, deep learning, logistic regression

## Abstract

Objective: The aim of this study was to develop, compare, and validate models for predicting cardiovascular disease (CVD) mortality and hospitalization with hypertension using a conventional statistical model and a deep learning model. Methods: Using the database of Korean National Health Insurance Service, 2,037,027 participants with hypertension were identified. Among them, CVD (myocardial infarction or stroke) death and/or hospitalization that occurred within one year after the last visit were analyzed. Oversampling was performed using the synthetic minority oversampling algorithm to resolve imbalances in the number of samples between case and control groups. The logistic regression method and deep neural network (DNN) method were used to train models for assessing the risk of mortality and hospitalization. Findings: Deep learning-based prediction model showed a higher performance in all datasets than the logistic regression model in predicting CVD hospitalization (accuracy, 0.863 vs. 0.655; F_1_-score, 0.854 vs. 0.656; AUC, 0.932 vs. 0.655) and CVD death (accuracy, 0.925 vs. 0.780; F_1_-score, 0.924 vs. 0.783; AUC, 0.979 vs. 0.780). Interpretation: The deep learning model could accurately predict CVD hospitalization and death within a year in patients with hypertension. The findings of this study could allow for prevention and monitoring by allocating resources to high-risk patients.

## 1. Introduction

Cardiovascular diseases (CVDs) such as coronary heart disease, stroke, and peripheral artery disease are well-known leading causes of mortality worldwide with approximately 18 million deaths in 2019, making it a global health burden [1,2,3]. Hypertension, one of the modifiable and preventable strong risk factors for CVDs, shows a rapidly increasing prevalence with more than 1.2 billion diagnosed with hypertension in 2019 [4,5,6]. Since lowering blood pressure is a promising method to reduce CVDs and related hospitalization and mortality [7,8], various types of antihypertensive drugs, surgical intervention, and nonpharmacological therapy have been advanced over the last decade [3]. Nevertheless, more than half of men and women with hypertension have not received suitable treatment, with few models that can predict the risk of complications [4]. Repeated hospitalization due to CVDs can reduce the quality of life, increase long-term mortality, and eventually increase socio-economic costs. To alleviate the burden of CVDs, the early detection or prediction of individuals with hypertension who are more likely to develop CVD related events is necessary for prevention and ti make appropriate decisions.

In order to improve the prognosis of patients, several CVD risk prediction models such as Framingham risk score (FRS) [9], thrombosis in myocardial infarction (TIMI) [10], systematic coronary risk evaluation (SCORE) [11], and QRISK [12] have been developed through regression-based methods in recent decades. These traditional statistical predictive models might be useful for an association analysis using a low number of variables. However, when a novel biomarker related to outcome is developed, it is necessary to analyze its accuracy in predicting outcome [13]. In addition, since large-scale, time-dependent datasets such as electronic health records (EHR) are available, an advanced model for maximizing risk stratification through repetitive measurements of explanatory variables and validation is needed [14].

Deep learning is a subset of machine learning that uses a multi-layered structure of algorithms called artificial neural networks (ANNs) inspired by human neural networks to perform automated feature learning [14]. To date, several studies in the cardiovascular field have shown the efficacy and potential of deep learning through high-accuracy disease prediction based on complex big data [15,16]. However, there is little research using deep learning models to predict prognosis related to CVD events in clinical patients with hypertension.

Thus, the aim of this study was to evaluate the discriminative accuracy of a deep learning-based prediction model for CVD mortality and hospitalization of patients with hypertension using the database of Korean National Health Insurance Service (NHIS) and to compare it with a conventional logistic regression model.

## 2. Materials and Methods

### 2.1. Study Design and Population

This study utilized the Korean National Health Information Database (NHID), a nationwide claims databases produced by the NHIS. Approximately 97% of Korean are enrolled in the NHIS. Mandatory health insurance is provided to enrolled citizens. The NHID includes sociodemographic data, lifestyle questionnaires, laboratory results, diagnoses, prescriptions, and data on death. Additional details about the database have been described elsewhere [17]. Diagnoses recorded in NHID were based on the International Classification of Diseases, Tenth Revision (ICD-10) codes.

The overall study design is shown in Figure 1. The cohort included 3,196,373 individuals not less than 19 years older years who were diagnosed with hypertension (I10-15 and O10-16) between 2002 and 2017. After excluding 159,346 individuals whose data on baseline variables were missing or who received two or more health checkups, 2,037,027 individuals were finally enrolled. Among them, we analyzed CVD-related death and/or hospitalization that occurred within one year after the last visit. CVD-related hospitalization was further subdivided into MI-related hospitalization (I21) and stroke-related hospitalization (I60-69).

This study was approved by Kangbuk Samsung Hospital Institutional Review Board. Informed consent was waived for this retrospective analysis. The authors are restricted from sharing the datasets underlying this study because the Korean NHIS owns the data. There are legal or ethical restrictions to sharing this data publicly. Data are available from the NHIS through a formal application process (https://nhiss.nhis.or.kr/bd/ab/bdaba021eng.do).

### 2.2. Study Variables

In this study, we analyzed data from the NHID to identify factors affecting CVD hospitalization and CVD death. We used the most recent age, sex, status of taking statin medications, and diabetes status for individuals who had the last visit with ICD-10 codes for hypertension (I10-15 and O10-16). Body mass index (BMI), systolic blood pressure (SBP), diastolic blood pressure (DBP), fasting plasma glucose (FPG), total cholesterol, income level, smoking status, and physical activity variables used in the national health screening database (DB) and eligibility DB were obtained. In addition, considering numerical change over time, we applied the value obtained by subtracting the last visit value from the first visit value of each variable. Change in the amount of smoking was defined as change in the number of cigarettes. Change in the amount of alcohol consumed per day and change in the amount of physical activity were defined as changes in results obtained using the formula according to the International Physical Activity Questionnaire (IPAQ) guidelines. To reflect information on history, the number of hospital visits, prescriptions, and length of hospitalization within the last two years from the last visit were utilized.

### 2.3. Clinical Outcomes

The primary outcome of this study was defined as CVD death (all I-code-related deaths in ICD-10), CVD hospitalization (all I-code-related hospitalizations in ICD-10), and a composite of CVD death and hospitalization within one year after the last visit.

### 2.4. Algorithm Development and Statistical Analysis

For statistical analysis, Chi-square test and independent sample *t*-test were conducted to compare differences according to CVD hospitalization, CVD death, and baseline characteristics using data from the NHIS. Multiple logistic regression analysis was performed to confirm the association between CVD hospitalization and CVD death for each variable. Logistic regression analysis and deep neural network (DNN) methods were used to build a predictive model. An overview of the data-processing of the DNN model is shown in Figure 2. Data-splitting was conducted at 7:3 for training and validation. Oversampling was performed using the synthetic minority oversampling technique (SMOTE) algorithm (because imbalanced data were used for CVD hospitalization and CVD death prediction) and min–max scaling (because units for each variable were different). The DNN model consisted of one hidden layer with 64 nodes. The Rectified Linear Unit (ReLU) activation function was used to apply nonlinearity in the hidden layer and the dropout ratio was set to 0.2 to prevent overfitting. To assess the predictive performance of the developed predictive model, we calculated their performance in terms of accuracy, precision (positive predictive value), recall (sensitivity), F_1_-score (harmonic mean between recall and precision), and area under ROC curve (AUC). The optimal cutoff was found using the Yuden index. All analyses were performed using SAS 9.4 program (SAS Institute Inc., Cary, NC, USA) and Python 3.7.0. All *p* values were two-tailed and *p* values less than 0.05 were defined as statistically significant.

## 3. Results

### 3.1. Patient Characteristics

The study analyzed a total of 2,037,027 patients with hypertension, including 163,686 participants with CVD hospitalization and 31,634 participants with CVD death. Baseline characteristics of participants are shown in Table 1. All variables showed statistically significant differences except for the variance of 0 income level for those with CVD hospitalization.

Participants with CVD hospitalization were older (66.3 ± 13.4 vs. 62.8 ± 12.9 years), more likely to be females (50.12 % vs. 49.88%), had a longer hospitalization period (32.22 ± 90.66 vs. 25.90 ± 69.61 days), higher proportion of history of diabetes (43.40% vs. 33.56%), higher proportion of myocardial infarction (MI) (2.16% vs. 0.87%), and higher proportion of history of stroke (12.52% vs. 3.98%). Meanwhile, they were less likely to take statins with decreased cholesterol level, reduced smoking, increased physical activity, increased prescriptions, and increased hospitalizations.

Participants with CVD death were older (75.9 ± 10.6 vs. 62.9 ± 12.9 years), more likely to be males (55.38 % vs. 51.37%), had more hospital visits (19.42 ± 20.67 vs. 13.34 ± 15.31), a longer hospitalization period (127.71 ± 196.46 vs. 24.81 ± 66.54 days), higher proportion of history of diabetes (49.42% vs. 34.11%), higher proportion of MI (9.62% vs. 0.84%), higher proportion of history of stroke (35.13% vs. 4.18%), and were more likely to have decreased total cholesterol level, but less likely to take statins with reduced smoking, increased physical activity, and increased prescriptions.

### 3.2. Variable Significance in Logistic Regression Model

Table 2 shows results of multiple regression analysis performed to evaluate the risk of variables affecting CVD hospitalization and CVD death using odds ratios (ORs) and 95% confidence intervals (CIs). Age, variance of BMI, variance of SBP, variance of FPG, variance of alcohol consumption, variance of physical activity, and history of diabetes increased the risk of CVD hospitalization and CVD death, irrespective of the presence of missing values. On the contrary, variance in DBP, number of prescriptions, and taking statins decreased the risk of CVD hospitalization and CVD death. The number of hospital visits decreased the risk of CVD hospitalization, but increased the risk of CVD death (OR [95% CI]: 0.995 [0.995, 0.995] vs. 1.005 [1.005, 1.005]).

### 3.3. Comparison of Model for Outcome Prediction

Table 3 shows the results after evaluating each model. For all metrics in all datasets, the DNN model showed a higher performance than the logistic regression model. In the case of ROC comparison, a statistically significant difference of *p* < 0.0001 was confirmed in all datasets. The DNN model showed a performance of 0.8 or higher for the CVD hospitalization group (accuracy, 0.863; F_1_-score, 0.854; precision, 0.912; recall, 0.803; AUC, 0.932) and 0.9 or higher for the CVD death group (accuracy, 0.925; F_1_-score, 0.924; precision, 0.935; recall, 0.912; AUC, 0.979) in all metrics. Figure 3 shows a comparison of ROC curves of a logistic regression model and DNN model for each dataset. Recall (sensitivity) was relatively lower across all subfigures in common, but it was all higher than recall in logistic regression (LR).

In a sub-analysis of CVD hospitalization, the DNN model showed a higher performance than the logistic regression model (Table 4, Figure 4). The DNN model showed an outstanding performance for both MI and stroke groups (AUC: 0.972 for MI and 0.951 for stroke). For both groups, all metrics of DNN model were higher than those of LR model within the test.

## 4. Discussion

In this retrospective cohort study based on a nationwide claim dataset, we developed a deep learning model to evaluate future risk of high-risk CVD mortality and hospitalization for patients with hypertension, and then compared the performance with that of a traditional statistical model. Among artificial intelligence studies using deep learning for the prediction of CVD-related outcomes in patients with pre-existing hypertension, ours is, to the best of our knowledge, the first conducted in a real-world setting to predict mortality and admission to hospital for hypertensive patients. We found that the prognostic efficacy of the deep learning method was higher than that of the logistic regression in all classification tasks using five commonly used evaluation metrics. In addition, the deep learning model showed excellent performance in sub-analysis performed by subcategorizing hospitalization due to MI and stroke.

Globally, the prevalence of hypertension in adults aged 30–79 is over 30%. The control rate of hypertension is only about 20%, despite the various mechanisms of hypertension that have been elucidated over a century, with effective drugs and interventions being developed [3,4,18]. Among the risk factors of CVD, the most important global burden on the health care system, hypertension is a crucial component in terms of its high prevalence and its being preventable cause of premature death worldwide [19]. However, there are not many risk assessment models or tools to predict the risk of CVD in hypertensive patients, and even tends to be overestimated in Korean and Asian populations because it is mainly considered in the United States (US) or European population [20].

Regression analysis is the most widely used analysis for medical data. Various models for predicting CVD in hypertensive patients have been proposed using regression analysis [21]. However, the predictors considered in the previous study [21] were only a few, traditional, strong risk factors that were already established. It is known that blood pressure control might have a limited effect on overall CVD risk due to the complexity and variability of individual cardiovascular risk. Therefore, novel risk factors related to CVD, including C-reactive protein (CRP), coronary artery calcium (CAC), and proprotein convertase subtilisin/kexin type 9 (PCSK9), have been proposed [22,23]. Although some newly discovered biomarkers have been confirmed to be associated with increased risk, a CVD prediction model using these markers failed to show superior performance to a traditional predictive model [24]. Logistic regression, used as a control in this study, is a conventional statistical approach frequently used to develop risk prediction models. The strength of this analysis lies in the determination and use of several variables to predict prognosis by expressing the predictive effect of predictor variables using simple and easy ways to explain parameters. Although a regression model with a low number of predictors, and the premise that the effect of the variable on the result is linear and homogeneous, can be useful in association analysis, it can act as a limitation in predictive analysis studies that focus on the result instead of the predictor variable [25]. Such a limitation cannot account for the multicollinearity or spatially varying correlations that may arise from the complexity and variability of an individual’s overall cardiovascular risk.

As large-scale cohort studies involving from tens to millions of participants have been established in various regions around the world, the accumulation of large datasets in the medical field has been accelerating as well. Analyses using these big data have provided great potential for research on complex problems beyond the scope of existing clinical and observational studies due to the vast amount of available data and ability to easily extract mortality data and perform disease registration. As a result, it is possible to analyze previously unknown risk factors that could have a statistically significant association with disease incidence using medical big data from a nationwide population-based cohort close to real world data, thereby tracking various disease mechanisms [26,27,28]. As an example, the incidence of CVD is influenced by, and often intertwined with, various risk factors such as race, ethnicity, age, sex, body mass index, and laboratory test results. Previous studies using a cohort with a limited population and traditional statistical analysis often could not fully reflect all the complex causal relationships between these various risk factors, leading to many limitations in the interpretation of results. Recently, the standardization and systematization of healthcare big data has enabled the analysis of previously unknown risk factors that might play a statistically significant role in disease prevalence, providing a good opportunity to more accurately characterize the quality of prevention and treatment according to CVD predictions [14]. The NHID used in this study is one of the largest claims datasets, with a population of over 52 million. It includes data for all age groups and all regions, thus reducing the possibility of selection bias [17]. In addition, it includes health-screening information including detailed lifestyle questionnaires, laboratory test results, and anthropometric measurements that are not included in other claim databases. Furthermore, since NHID is linked to other data, such as mortality data from the National Statistical Office, it enables a more detailed analysis.

On the other hand, machine learning has been introduced to overcome the limitations of risk prediction using traditional regression analysis [25]. In particular, research using artificial intelligence (AI) and big data has recently attracted attention and disease prediction research using AI has already shown a high value in other studies [29,30,31]. Unsupervised deep learning models are trained either consciously or unconsciously by updating and adjusting neuron weights and biases as relevant knowledge is identified. Thus, they can detect patterns or potential risk factors that humans cannot detect, utilizing more computational power to handle all possible variables. The DNN model utilized in this study had a relatively simple structure with one hidden layer. Nevertheless, the strength of our study is that if variables are used in the analysis of this study, it will be possible to find people who are highly likely to be hospitalized within one year due to CVD, enabling an immediate response for prevention that is not possible with existing predictive models. In future studies, we will investigate interpretable and complicated structures of DNN models with multiple hidden layers and evaluate other novel deep learning models.

## 5. Study Limitations

The results of our study should be interpreted in the context of several potential limitations. First, in this study, hypertensive patients were diagnosed based on the ICD-code with potential errors in the claims data, not by medical records or anthropometric measurements, as in other large-data studies. However, since previous studies reported that ICD-code showed acceptable reliability in the NHID in Korea [32], this selection method was considered reasonable.

Second, external evaluation was not conducted to determine the reproducibility or generalizability. Although we validated a single cohort by splitting it into development and validation datasets, the accuracy of prediction might have been lowered when comparing with cohorts with different regions, races, countries, and other types of care settings. However, in another study that validated predictive models using NHIS for Koreans through Rotterdam studies for European, the deep learning approach showed significantly improved predictive power compared to the approach through traditional regression analysis, suggesting that it could be generalized without geographic or racial influence [29]. Finally, data analysis using deep learning models can have difficulies in accurately understanding and explaining individual algorithms inside the model. The complexity of the deep learning model enables a distinguished prediction power. At the same time, it makes it more difficult to understand and trust. Furthermore, it is not possible to provide specific recommendations for controlling risk factors because the risk factors affecting event occurrence are unknown. Therefore, to find ways to explain individual risk factors, it is necessary to train models using both interpretable and deep learning methods to evaluate whether there is a trade-off between accuracy and interpretability in a specific case in practice.

## 6. Conclusions

Using experimental machine learning techniques based on high-quality big data, we showed that a machine learning approach using DNN could lead to more accurate predictions of the risk of mortality and hospitalization in patients with hypertension than a conventional statistical approach. Further research should be conducted by identifying patients at high risk of hospitalization or death from CVD after being diagnosed with hypertension through prediction models developed using various deep-learning-based techniques so that customized treatment and monitoring can be achieved by allocating resources to the patients at highest risk.

## Figures and Tables

**Figure 1 jcm-11-06677-f001:**
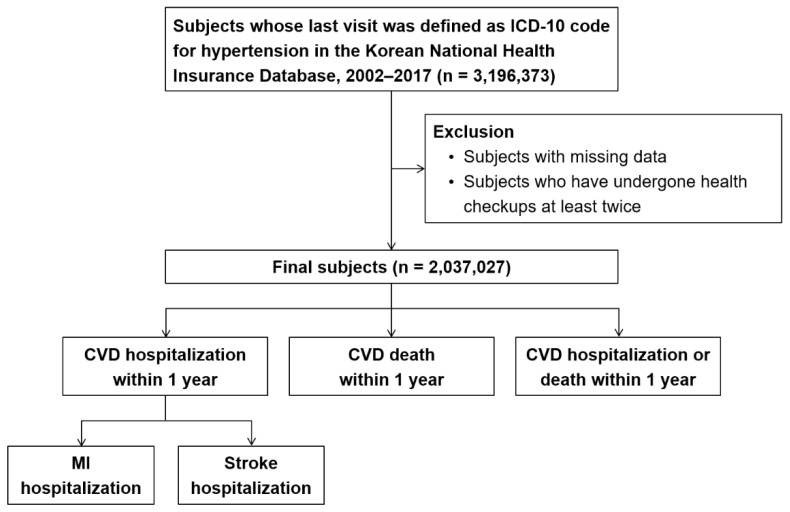
Flowchart showing the selection of study population. CVD: cardiovascular disease. ICD: international classification of diseases. MI: myocardial infarction.

**Figure 2 jcm-11-06677-f002:**
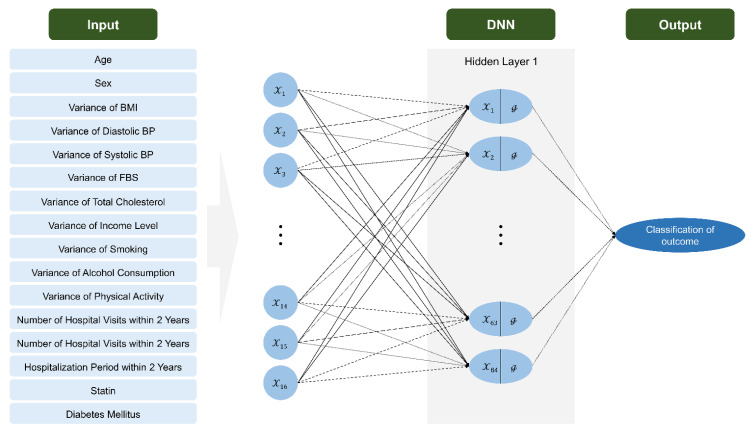
The structure of DNN model for outcome classification. Circles represent neurons. The arrows indicate heavy-weight transmissions between neurons and dashed arrows indicate the invalid neuronal connection. BMI: body mass index. BP: blood pressure. DNN: deep neural network.

**Figure 3 jcm-11-06677-f003:**
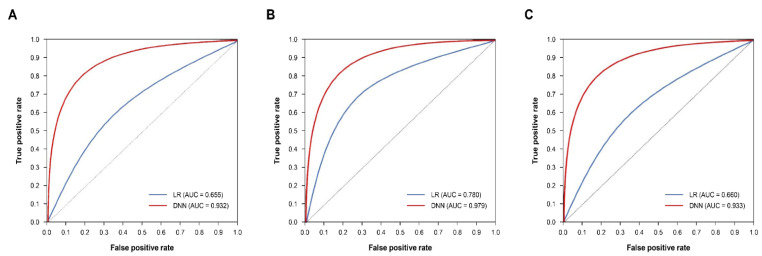
Comparison of ROC curves. (**A**) CVD hospitalization, (**B**) CVD death, (**C**) CVD total (hospitalization + death). (**A**) MI, (**B**) Stroke. AUC: area under the curve. DNN: deep neural network. LR: logistic regression. MI: myocardial infarction. ROC: receiver operating characteristic. *p* < 0.0001 for all comparisons of ROC curves.

**Figure 4 jcm-11-06677-f004:**
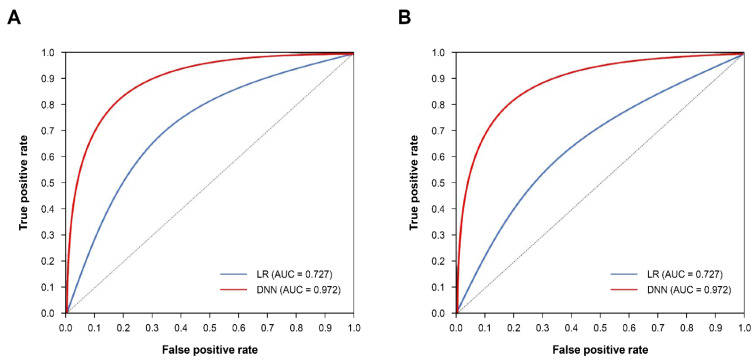
Comparison of ROC curves. (**A**) MI, (**B**) Stroke. AUC; area under the curve. DNN: deep neural network. LR: logistic regression. MI: myocardial infarction. ROC: receiver operating characteristic.

**Table 1 jcm-11-06677-t001:** Baseline characteristics of study participants in CVD hospitalization and CVD death groups.

Variables	CVD Hospitalization			CVD Death		
	No (n = 1,873,341)	Yes (n = 163,686)	*p* Value	No (n = 2,005,393)	Yes (n = 31,634)	*p* Value
Age, years	62.8 ± 12.9	66.3 ± 13.4	<0.0001	62.9 ± 12.9	75.9 ± 10.6	<0.0001
Sex			<0.0001			<0.0001
Male	966,057 (51.57)	81,654 (49.88)		1,030,193 (51.37)	17,518 (55.38)	
Female	907,284 (48.43)	82,032 (50.12)		975,200 (48.63)	14,116 (44.62)	
Variance of BMI	−0.17 ± 8.69	0.12 ± 7.27	<0.0001	−0.16 ± 8.61	0.64 ± 6.85	<0.0001
Variance of Diastolic BP	2.39 ± 13.72	1.57 ± 13.16	<0.0001	2.31 ± 13.65	2.90 (15.26)	<0.0001
Variance of Systolic BP	1.11 ± 20.20	0.16 ± 19.63	<0.0001	0.99 ± 20.09	4.04 (24.07)	<0.0001
Variance of FBS	−6.82 ± 35.93	−4.78 ± 37.63	<0.0001	−6.73 ± 35.87	−1.83 (46.87)	<0.0001
Variance of total cholesterol	9.96 ± 56.37	5.83 ± 54.77	<0.0001	9.61 ± 56.16	10.78 (61.97)	0.0009
Variance of income level	−82,776.98 ± 104,588.08	−82,947.07 ± 105,521.46	0.5314	−82,975.97 ± 104,855.49	−71,041.90 ± 90,900.10	<0.0001
Variance of smoking	1.32 ± 6.89	1.21 ± 6.66	<0.0001	1.32 ± 6.87	0.91 ± 6.47	<0.0001
Variance of alcohol consumption	−25.86 ± 105.35	−17.19 ± 98.33	<0.0001	−25.38 ± 105.15	−11.87 ± 80.59	<0.0001
Variance of physical activity	−0.62 ± 3.01	−0.42 ± 3.00	<0.0001	−0.61 ± 3.01	0.08 ± 2.80	<0.0001
Number of hospital visits within 2 years	13.73 ± 15.22	10.11 (17.29)	<0.0001	13.34 ± 15.31	19.42 ± 20.67	<0.0001
Number of prescriptions within 2 years	462.48 ± 343.64	290.77 ± 346.72	<0.0001	449.01 ± 347.20	427.84 ± 335.88	<0.0001
Hospitalization period within 2 years, days	25.90 ± 69.61	32.22 ± 90.66	<0.0001	24.81 ± 66.54	127.71 ± 196.46	<0.0001
Statin			<0.0001			<0.0001
No	1,156,157 (61.72)	117,151 (71.57)		1,248,568 (62.26)	24,740 (78.21)	
Yes	717,184 (38.28)	46,535 (28.43)		756,825 (37.74)	6,894 (21.79)	
Diabetes			<0.0001			<0.0001
No	1,244,737 (66.44)	92,650 (56.60)		1,321,387 (65.89)	16,000 (50.58)	
Yes	628,604 (33.56)	71,036 (43.40)		684,006 (34.11)	15,634 (49.42)	
MI			<0.0001			<0.0001
No	1,857,062 (99.13)	160,155 (97.84)		1,988,626 (99.16)	28,592(90.38)	
Yes	16,279 (0.87)	3,531 (2.16)		16,767 (0.84)	3,043(9.62)	
Stroke			<0.0001			<0.0001
No	1,798,818 (96.02)	143,190 (87.48)		1,921,488 (95.82)	20,520 (64.87)	
yes	74,523 (3.98)	20,496 (12.52)		83,905 (4.18)	11,114 (35.13)	

BMI: body mass index, BP: blood pressure, CVD: cardiovascular disease, FBS: fasting blood sugar, MI: myocardial infarction.

**Table 2 jcm-11-06677-t002:** Results of multiple logistic regression.

Variable	CVD Hospital		CVD Death		CVD Total (Hospital + Death)	
	Missing Value (+) OR (95% CI)	Missing Value (−) OR (95% CI)	Missing Value (+) OR (95% CI)	Missing Value (−) OR (95% CI)	Missing Value (+) or (95% CI)	Missing Value (−) or (95% CI)
Age, years	1.028 (1.027-1.028)	1.021 (1.021–1.021)	1.094 (1.092–1.095)	1.074 (1.074–1.075)	1.035 (1.035–1.036)	1.035 (1.035–1.036)
Sex						
Male	Reference	Reference	Reference	Reference	Reference	Reference
Female	1.016 (1.005–1.027)	0.973 (0.965–0.981)	0.640 (0.625–0.655)	0.815 (0.806–0.825)	0.961 (0.951–0.971)	0.947 (0.940–0.954)
Variance of BMI	1.001 (1.000–1.000)	1.001 (1.001–1.001)	1.001 (1.001–1.002)	1.001 (1.000–1.001)	1.001 (1.001–1.001)	1.001 (1.001–1.001)
Variance of Diastolic BP	0.998 (0.998–0.999)	0.997 (0.997–0.998)	0.989 (0.988–0.990)	0.985 (0.985–0.986)	0.996 (0.996–0.997)	0.994 (0.993–0.994)
Variance of Systolic BP	1.001 (1.001–1.001)	1.001 (1.001–1.002)	1.009 (1.008–1.010)	1.008 (1.007–1.008)	1.003 (1.002–1.003)	1.003 (1.003–1.004)
Variance of FBS	1.001 (1.001–1.001)	1.001 (1.001–1.002)	1.002 (1.001–1.002)	1.003 (1.002–1.003)	1.001 (1.001–1.001)	1.002 (1.002–1.002)
Variance of Total Cholesterol	0.999 (0.999–1.000)	0.999 (0.999–0.999)	1.000 (0.999–1.000)	0.998 (0.998–0.999)	0.999 (0.999–0.999)	0.999 (0.999–0.999)
Variance of Income level	1.000 (1.000–1.000)	1.000 (1.000–1.000)	1.000 (1.000–1.000)	1.000 (1.000–1.000)	1.000 (1.000–1.000)	1.000 (1.000–1.000)
Variance of smoking	1.000 (0.999–1.001)	0.997 (0.997–0.998)	0.992 (0.990–0.994)	0.979 (0.977–0.980)	0.999 (0.998–1.000)	0.993 (0.993–0.994)
Variance of alcohol consumption	1.001 (1.000–1.001)	1.001 (1.001–1.001)	1.001 (1.000–1.001)	1.001 (1.001–1.001)	1.001 (1.001–1.001)	1.001 (1.001–1.001)
Variance of Physical activity	1.009 (1.007–1.011)	1.013 (1.011–1.015)	1.030 (1.026–1.034)	1.035 (1.032–1.037)	1.012 (1.011–1.014)	1.019 (1.017–1.020)
Number of hospital visits within 2 years	0.991 (0.991–0.992)	0.995 (0.995–0.995)	1.007 (1.006–1.007)	1.005 (1.005–1.005)	0.997 (0.997–0.998)	1.000 (1.000–1.000)
Number of prescriptions within 2 years	0.998 (0.998–0.998)	0.999 (0.999–0.999)	0.999 (0.999–0.999)	0.999 (0.999–0.999)	0.998 (0.998–0.998)	0.999 (0.999–0.999)
Hospitalization period within 2 years, days	1.000 (1.000–1.000)	1.000 (1.000–1.000)	1.003 (1.003–1.003)	1.002 (1.002–1.002)	1.001 (1.001–1.001)	1.001 (1.00–1.001)
Statin						
No	Reference	Reference	Reference	Reference	Reference	Reference
Yes	0.640 (0.632–0.647)	0.648 (0.642–0.654)	0.534 (0.519–0.549)	0.489 (0.481–0.497)	0.617 (0.610–0.624)	0.584 (0.579–0.589)
Diabetes						
No	Reference	Reference	Reference	Reference	Reference	Reference
Yes	1.749 (1.730–1.768)	1.880 (1.866–1.896)	1.657 (1.619–1.696)	1.384 (1.368–1.401)	1.715 (1.698–1.733)	1.708 (1.696–1.721)

BMI: body mass index, BP: blood pressure, CVD: cardiovascular disease, FBS: fasting blood sugar, MI: myocardial infarction.

**Table 3 jcm-11-06677-t003:** Performances of logistic regression and deep neural network models for main outcomes.

Variable	CVD Hospital				CVD Death				CVD Total (Hospital + Death)			
	Missing Value (+)		Missing Value (−)		Missing Value (+)		Missing Value (−)		Missing Value (+)		Missing Value (−)	
	LR	DNN	LR	DNN	LR	DNN	LR	DNN	LR	DNN	LR	DNN
Accuracy	0.646	0.824	0.655	0.863	0.777	0.886	0.780	0.925	0.673	0.843	0.659	0.860
F1 score	0.648	0.811	0.656	0.854	0.782	0.889	0.783	0.924	0.672	0.834	0.658	0.852
Precision	0.645	0.877	0.654	0.912	0.764	0.870	0.773	0.935	0.675	0.879	0.662	0.903
Recall	0.650	0.754	0.658	0.803	0.800	0.908	0.793	0.912	0.669	0.794	0.654	0.807
AUC	0.646	0.907	0.655	0.932	0.777	0.959	0.780	0.979	0.673	0.923	0.660	0.933

CVD: cardiovascular disease, DNN: deep neural network, LR: logistic regression.

**Table 4 jcm-11-06677-t004:** Performances of logistic regression and deep neural network models for MI and stroke.

Variable	MI				Stroke			
	Missing Value (+)		Missing Value (−)		Missing Value (+)		Missing Value (−)	
	LR	DNN	LR	DNN	LR	DNN	LR	DNN
Accuracy	0.686	0.911	0.727	0.917	0.681	0.883	0.682	0.897
F1 score	0.696	0.908	0.734	0.916	0.687	0.874	0.690	0.889
Precision	0.675	0.939	0.715	0.927	0.674	0.935	0.674	0.956
Recall	0.718	0.879	0.754	0.907	0.700	0.824	0.707	0.832
AUC	0.686	0.968	0.727	0.972	0.681	0.948	0.682	0.951

AUC: Area under the curve, DNN: deep neural network, LR: logistic regression, MI: myocardial infarction. *p* < 0.0001 for all two groups.

## Data Availability

Not applicable.

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
