# Peer review of "Deep Learning Improves Prediction of Cardiovascular Disease-Related Mortality and Admission in Patients with Hypertension: Analysis of the Korean National Health Information Database"

_jcm, 2022, doi:10.3390/jcm11226677_

Round 1
Reviewer 1 Report
In this manuscript, the authors present a retrospective cohort study based on a nationwide claim dataset. The authors proposed a deep learning model to evaluate future risk of high-risk cardiovascular diseases (CVD) mortality and hospitalization for patients with hypertension and then compared the performance to a canonical statistical model. The proposal outreached the results obtained by other AI-based studies using deep learning for the prediction of CVD-related outcomes in patients with pre-existing hypertension. According to the authors, their proposal was the first one conducted in a real-world setting to predict mortality and admission to hospital for hypertensive patients. The authors proposal surpassed the logistic regression in all classification tasks using accuracy, precision, recall, F1-score, and AUC. The proposed model also reached high accuracy results for sub-analysis performed by sub-categorizing hospitalization due to myocardial infarction (MI) and stroke. The manuscript is well-written and organized. The theoretical foundations are strong. The methodological aspects are clear and seem reproducible. The conclusion is quit rich and sufficiently profound. I just suggest the authors to include sensitivity and specificity. Please also include boxplots for sensitivity, specificity and AUC as well, and adjust the discussion section accordingly.
Author Response
Thank you for your valuable suggestion.
This data was applied to the National Health Insurance Corporation. Unfortunately, the data could not be checked after the data usage period is over. Although it is impossible to create sensitivity, specificity, and AUC through boxplot, it is judged that it is possible to make inferences using the values of Precision, Recall, and Accuracy presented by us.

Reviewer 2 Report
The authors utilized the logistic regression method and deep neural network (DNN) method to predict cardiovascular disease (CVD) mortality and hospitalization with hypertension in a large patients’ population with 2,037,027 participants. The results indicated that deep learning was more accurate in predicting cardiovascular disease (CVD) mortality and hospitalization than logistic regression method. These results may allow identifying high-risk patients and allocating resources to them in terms of prevention and monitoring.
The other researchers used the regression-based methods to build the risk prediction model. However, few researches used deep learning models to predict prognosis related to CVD events in clinical patients. This was the first research conducted in a real-world setting to predict mortality and admission to hospital for hypertensive patients. The writing is overall fluent and accurate. The statical methods were proper and valid. There were a few advice for the authors before the publication of this work.
1、 In Table 3, ROC comparison, a statistically significant difference of p<0.0001 was confirmed in all datasets. In figure 3, the p value could better be marked in the figure, so that the readers could more easily interpretate the figures.
2、 In Table 4, were there statistically significant difference between the AUCs from the two groups?
3、 In line 123, Data splitting was conducted at 7:3 for training and validation. There seems to be no results relevant to training and validation stated in the results sessions.
4、 In line 127, the dot in the p value (0.0001) was not the correct format.
5、 Typically, the logistic regression was performed with two steps. First, uni-variate logistic regression was performed to identify variables significantly correlated with the outcome. Second, multi-variate logistic regression was used to build the prediction model and to identify the most correlated variables. Could the DNN model identify the key features contributing the prediction model? Were the key variables identified by the two methods different to each other?
6、 This work laid a foundation for using DNN in risk prediction of hypertension in terms of comparing with traditional logistic regression method. In the future study, the authors could develop a prediction system for single patient risk stratification so that this system could be used in patient counselling.
Author Response
- In Table 3, ROC comparison, a statistically significant difference of p<0.0001 was confirmed in all datasets. In figure 3, the p value could better be marked in the figure, so that the readers could more easily interpretate the figures.
Thank you for your valuable comment.
We added the following sentence to the footnote of Figure 3.
“p < 0.0001 for all comparisons of ROC curves.”
- In Table 4, were there statistically significant difference between the AUCs from the two groups?
Thank you for your valuable comment.
We added the following sentence to the footnote of Figure 3.
“p < 0.0001 for all two groups”
- In line 123, Data splitting was conducted at 7:3 for training and validation. There seems to be no results relevant to training and validation stated in the results sessions.
Thanks for pointing that out.
We split our dataset 7:3. 70% was used for training and 30% for validation. Table 2, Table 3, Table 4, Figure 3, and 4 presented by us are the results derived from the test set.
- In line 127, the dot in the p value (0.0001) was not the correct format.
Thank you for your valuable comment.
We corrected the format as follows.
Before revision
p<0∙0001
After revision
p<0.0001
- Typically, the logistic regression was performed with two steps. First, univariate logistic regression was performed to identify variables significantly correlated with the outcome. Second, multivariate logistic regression was used to build the prediction model and to identify the most correlated variables. Could the DNN model identify the key features contributing the prediction model? Were the key variables identified by the two methods different to each other?
Thank you for your valuable comment.
The logistic regression coefficients and the DNN model's variable importance were extracted and compared. The importance of the top five variables was identified, and the results were shown in the table below. Since the list of variable importance extracted from the model was similar depending on whether there is a missing value, the table below presents the result when the value is Missing (-). It could be seen that the variable importance extracted according to the two models is almost the same, although the magnitudes may be slightly different.
|
Outcome |
Classifier |
Variable importance |
|
CVD hospital |
LR |
Diabetes > Statin > Sex > Age > Number of hospital visits within 2years |
|
DNN |
Sex > Diabetes > Statin > Number of hospital visits within 2years > Number of prescriptions within 2years |
|
|
CVD death |
LR |
Statin > Diabetes > Sex > Age > Variance of Physical activity |
|
DNN |
Statin > Diabetes > Sex > Hospitalization period within 2 years > age |
|
|
CVD total |
LR |
Diabetes > Statin > Sex > Age > Variance of Physical activity |
|
DNN |
Sex > Diabetes > Visit > Statin > age |
|
|
MI |
LR |
Diabetes > Statin > Sex > Age > Number of hospital visits within 2years |
|
DNN |
Sex > Statin > Diabetes > Number of hospital visits within 2years > Number of prescriptions within 2years |
|
|
Stroke |
LR |
Statin > Diabetes > Sex > Age > Number of hospital visits within 2years |
|
DNN |
Statin > Diabetes > Sex > Age > Number of hospital visits within 2years |
- This work laid a foundation for using DNN in risk prediction of hypertension in terms of comparing with traditional logistic regression method. In the future study, the authors could develop a prediction system for single patient risk stratification so that this system could be used in patient counselling.
Thank you for your valuable comment.
We will reflect your advice in future research.

Round 2
Reviewer 2 Report
The revised manuscript has addressed all my comments.
Minor:
1.Figure legend of Fig 2 needs more details.
2. Please go through the text to remedy spelling and grammar issues.
Author Response
1.Figure legend of Fig 2 needs more details.
Thank you for your valuable comment.
We added the some sentences to the legend of Fig 2
Before revision
DNN model for outcome classification. BMI: body mass index. BP: blood pressure. DNN: deep neural network.
After revision
The structure of DNN model for outcome classification. Circles represent neurons. The arrows indicate heavy-weight transmissions between neurons and dashed arrows indicate the invalid neuronal connection. BMI: body mass index. BP: blood pressure. DNN: deep neural network
2. Please go through the text to remedy spelling and grammar issues.
Thank you for your valuable comment.
We reviewed the text again and corrected the problematic part.
